# The evolutionary history of the ancient weevil family Belidae (Coleoptera: Curculionoidea) reveals the marks of Gondwana breakup and major floristic turnovers, including the rise of angiosperms

Xuankun Li[1,2,3]*, Adriana E Marvaldi[4,5]*, Rolf G Oberprieler[6], Dave Clarke[2,3], Brian D Farrell[7], Andrea Sequeira[8], M Silvia Ferrer[5], Charles O'Brien[9†], Shayla Salzman[10], Seunggwan Shin[11], William Tang[12], Duane D McKenna[2,3]

[1]Department of Entomology, College of Plant Protection, China Agricultural University, Beijing, China; [2]Department of Biological Sciences, University of Memphis, Memphis, United States; [3]Center for Biodiversity Research, University of Memphis, Memphis, United States; [4]CONICET, División Entomología, Facultad de Ciencias Naturales y Museo, Universidad Nacional de La Plata, Buenos Aires, Argentina; [5]CONICET, Instituto Argentino de Investigaciones de Zonas Áridas, Entomología, Mendoza, Argentina; [6]CSIRO, Australian National Insect Collection, Canberra, Australian Capital Territory, Canberra, Australia; [7]Department of Organismic and Evolutionary Biology, Harvard University, Cambridge, United States; [8]Department of Biological Sciences, Wellesley College, Wellesley, United States; [9]West Calle Balaustre, Green Valley, Green Valley, United States; [10]Department of Entomology, University of Georgia, Athens, United States; [11]School of Biological Sciences, Seoul National University, Seoul, Republic of Korea; [12]Florida State Collection of Arthropods, Florida Department of Agriculture – DPI, Gainesville, United States

*For correspondence:
xuankun.li@cau.edu.cn (XL);
marvaldi@fcnym.unlp.edu.ar
(AEM)

†Deceased

## eLife assessment

Through anchored phylogenomic analyses, this **important** study offers fresh insights into the evolutionary history of the plant diet and geographic distribution of Belidae weevil beetles. Employing robust methodological approaches, the authors propose that certain belid lineages have maintained a continuous association with Araucaria hosts since the Mesozoic era. Although the biogeographical analysis is somewhat limited by uncertainties in vicariance explanations, this **convincing** study enhances our understanding of Belidae's evolutionary dynamics and provides new perspectives on ancient community ecology.

**Abstract** The rise of angiosperms to ecological dominance and the breakup of Gondwana during the Mesozoic marked major transitions in the evolutionary history of insect-plant interactions. To elucidate how contemporary trophic interactions were influenced by host plant shifts and palaeogeographical events, we integrated molecular data with information from the fossil record to construct a time tree for ancient phytophagous weevils of the beetle family Belidae. Our analyses

indicate that crown-group Belidae originated approximately 138 Ma ago in Gondwana, associated with Pinopsida (conifer) host plants, with larvae likely developing in dead/decaying branches. Belids tracked their host plants as major plate movements occurred during Gondwana's breakup, surviving on distant, disjunct landmasses. Some belids shifted to Angiospermae and Cycadopsida when and where conifers declined, evolving new trophic interactions, including brood-pollination mutualisms with cycads and associations with achlorophyllous parasitic angiosperms. Extant radiations of belids in the genera *Rhinotia* (Australian region) and *Proterhinus* (Hawaiian Islands) have relatively recent origins.

## Introduction

The evolutionary interplay between plant-feeding (phytophagous) insects and vascular plants has fundamentally shaped terrestrial biodiversity for over 400 million years (e.g. *Ehrlich and Raven, 1964*; *Farrell et al., 1992*; *Wilf, 2008*; *Misof et al., 2014*; *Swain et al., 2022*). The fossil record has provided insights into the historical context and evolutionary processes that have shaped modern insect-plant interactions (*Labandeira and Currano, 2013*). However, (1) the degree to which contemporary trophic interactions reflect those in the past, e.g., representing relatively ancient primary associations versus more recent secondary associations, and (2) the relative roles of vicariance (including major palaeo-geographical events) and dispersal in the evolution of host shifts, remain unclear.

Gymnosperms (cycads, conifers and relatives), the dominant plant group during the Mesozoic Era, thrived from the Permian to the Cretaceous period, dominating most terrestrial ecosystems and forming intimate associations with many groups of phytophagous insects (*Anderson et al., 2007*). The transition to the Cretaceous marked the rise of angiosperms (flowering plants), which gradually replaced gymnosperms in many ecological settings to become the dominant plant group (*Anderson et al., 2007*). This transition was not only a botanical shift but also marks a critical period in the evolutionary history of insect-plant interactions. Angiosperms introduced novel ecological niches and resources, leading to the diversification of phytophagous insects and a reconfiguration of insect-plant associations (e.g. *Farrell, 1998*; *Wilf, 2008*; *McKenna et al., 2009*; *Mckenna et al., 2015*; *Swain et al., 2022*). By the end of the Mesozoic, the continents had rifted into nearly their present forms, though their positions would continue to change. Gondwana split into South America, Africa, Australia, Antarctica and the Indian subcontinent, while Laurasia became North America and Eurasia.

On account of their extraordinary taxonomic diversity and varied trophic interactions with contemporary gymnosperms and flowering plants, beetles (order Coleoptera; >400,000 described extant species) have been widely used as models to study the evolution of diversity at the insect-plant interface (e.g. *Farrell, 1998*; *McKenna et al., 2009*; *McKenna et al., 2019*). The present paper addresses a significant gap in our understanding of the biogeography and evolution of host plant associations in the family Belidae (belid weevils), an ancient group of phytophagous beetles whose extant species exhibit specialized trophic associations with gymnosperms and flowering plants. By reconstructing the evolutionary history of belid weevils and examining the dynamics of their host plant associations and geographical distributions over time, this paper seeks to yield new insights into the evolution of modern insect-plant interactions.

Belidae, with approximately 360 described extant species in 40 genera and two subfamilies (*Marvaldi and Ferrer, 2014*; *O'Brien and Tang, 2015*), comprise an early branch of Curculionoidea (*Oberprieler et al., 2007*; *McKenna et al., 2009*; *McKenna et al., 2018*; *McKenna, 2011*; *Figure 1*). Belidae are indicated to have originated during the Jurassic in association with gymnosperms and subsequently colonized angiosperms, as the latter diversified and rose to ecological dominance during the Late Cretaceous and Paleogene (e.g. *Farrell, 1998*; *McKenna et al., 2009*; *Shin et al., 2018*). They exhibit their highest generic diversity in the Southern Hemisphere and collectively are associated with a variety of plants, including conifers, cycads and a few families of angiosperms, including Arecaceae, Balanophoraceae, Hydnoraceae, Celastraceae, Myrtaceae and Vitaceae (*Marvaldi and Ferrer, 2014*). The adults appear to feed mainly on stem tissues but sometimes also on pollen, some being important pollinators, and the larvae develop in the bark and woody tissues of decaying branches or twigs, gymnosperm strobili, flower buds or fruits. Neotropical Belidae in the subtribe Allocorynina develop as brood pollination mutualists in the pollen cones (or 'male strobili') of cycads (e.g. *Tang,*

**eLife digest** For over 400 million years, insects and plants have evolved alongside one another, shaping each other's development while adapting to major environmental changes. Studying fossils can provide clues about how ancient interactions between plants and insects developed as well as how environmental changes influenced these relationships.

During the Mesozoic era (around 252 to 66 million years ago) two events led to major changes in how insects and plants interacted. Firstly, flowering plants began to replace gymnosperms – plants whose seeds are not enclosed in a protective case – such as conifers, which were dominant at the time. Secondly, the supercontinent Gondwana split into the separate land masses of today, including South America, Africa, Antarctica and Australia.

To understand how these changes affected insect-plant relationships, Li et al. studied the evolutionary history of the Belidae family of beetles. During the Jurassic period (around 200-150 million years ago) these beetles lived off gymnosperms, but later they began to feed on flowering plants.

By combining genetic information from the DNA of 38 species of Belidae beetles with information from fossils, Li et al. constructed a timeline of the beetles' evolution. This revealed that Belidae beetles first appeared around 138 million years ago in Gondwana. Their larvae likely developed in the dead and decaying branches of the conifer plants on which they fed. As Gondwana split, these insects remained with their conifer hosts on the newly formed land masses. However, when conifers became less common, some of the beetles switched to feeding and developing larvae on flowering plants instead, diversifying again.

Understanding how changes in the availability of plants and the Earth's geography have affected insects in the past can help scientists understand evolutionary history as well as current ecosystem stability and biodiversity. With further research, this may help scientists to devise strategies to better manage and preserve ecosystems.

*1987*; *Marvaldi and Ferrer, 2014*; *Salzman et al., 2020*), a biotic interaction that has several independent origins in weevils.

The historical biogeography and evolution of host associations of Belidae have long drawn the attention of scientists (*Kuschel, 1959*; *Anderson, 2005*; *Marvaldi et al., 2006*). However, our understanding of belid evolution remains limited due to the lack of dated phylogeny estimates. Since the landmark study of weevil family-level phylogeny by *Kuschel, 1995*, several studies based on morphological characters of adults and larvae have inferred the phylogenetic relationships of Belidae and revised their tribal-level classification (*Kuschel and Leschen, 2003*; *Anderson, 2005*; *Marvaldi, 2005*; *Marvaldi et al., 2006*), and hypotheses for generic phylogenetic relationships based on analyses of morphological data have been proposed for both the subfamilies Belinae (*Kuschel and Leschen, 2003*) and Oxycoryninae (*Marvaldi et al., 2006*; *Anderson and Marvaldi, 2013*). However, beyond including small numbers of exemplar taxa in higher-level studies of weevils (e.g. *McKenna et al., 2009*; *Shin et al., 2018*), the phylogeny and evolution of Belidae have not been explored using molecular data.

We conducted molecular phylogenetic analyses and divergence dating of belid weevils to investigate their relationships and evolution. We also reconstructed ancestral states of host plant associations and undertook a biogeographical analysis to explore geographical patterns of diversification and the evolution of host plant organ and taxon associations. We integrated phylogenomic and Sanger data for Belidae, sampling all seven tribes and 60% of the extant genera (Combining Sanger sequences data with genomic data for phylogenetic inference has been demonstrated as a feasible approach to resolving deep-level relationships while adding taxa to the phylogeny for tracing the evolutionary history of characters e.g., *Zhang et al., 2016*; *Song et al., 2020*; *Li et al., 2022a*; *Li et al., 2022b*). We performed ancestral-state reconstruction using the resulting chronogram with stochastic character mapping and event-based likelihood ancestral-area estimation. We sought to answer two main questions about the distribution and host plant associations of crown group Belidae: (1) is the development of larvae of Agnesiotidini and Pachyurini (Belinae) and Oxycraspedina (Oxycoryninae) in conifer hosts an ancient, primary association (*Farrell, 1998*) or the result of more recent, secondary colonization, and (2) how did the interplay between biogeographical process and host plant shifts influence and shape the trophic associations of belids?

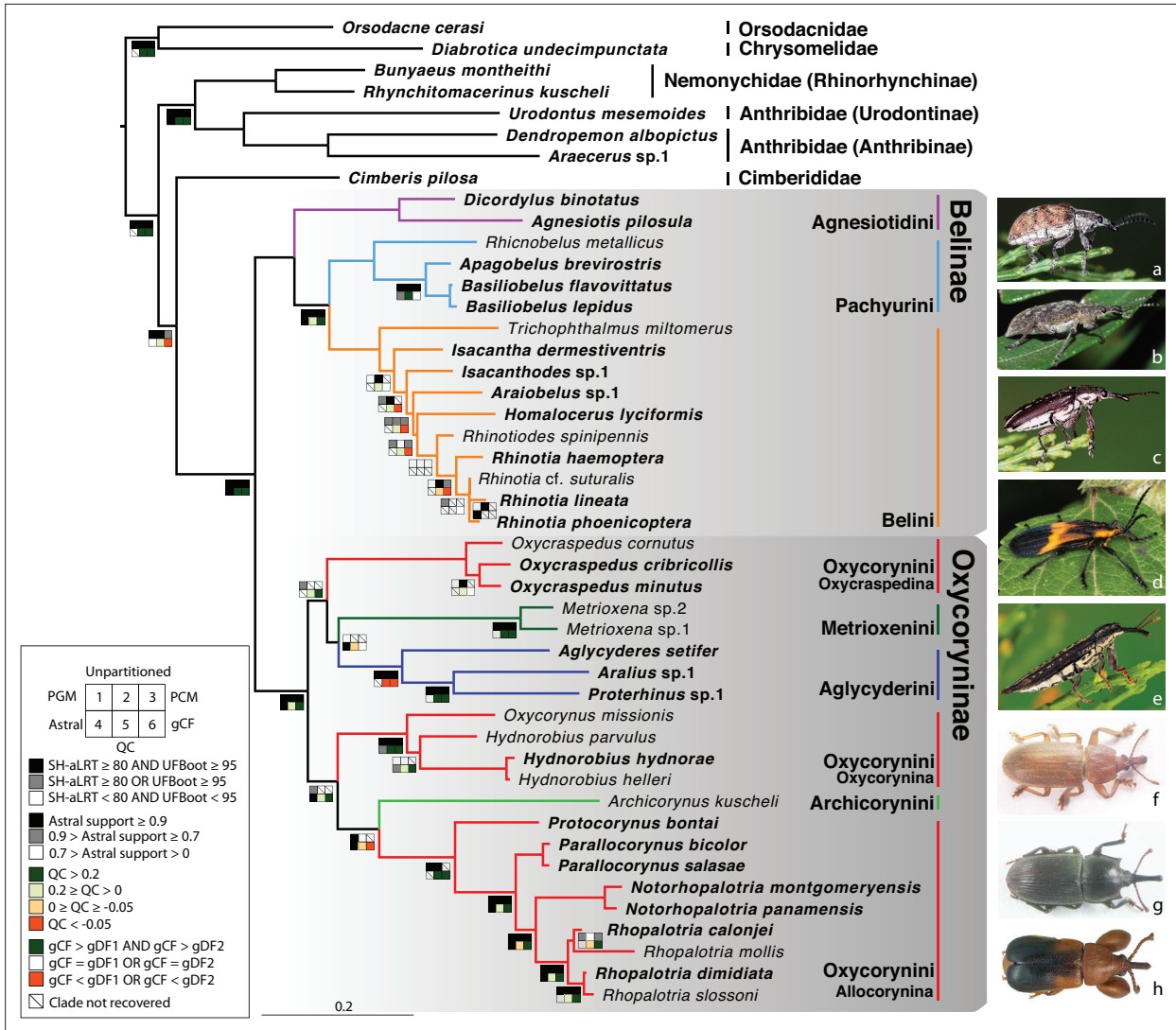

**Figure 1.** Phylogeny estimate of Belidae. Topology obtained from analysis of the 46t424g matrix via maximum likelihood (ML), partitioned by gene. Squares along branches are ML SH-aLRT and UFBoot values for 1–3: 1. partition by gene, 2. unpartitioned, and 3. partition by gene and codon positions. Astral support for 4, Quartet Concordance (QC) for 5, and gene concordance factor (gCF) for 6. Branches without squares indicate the highest support in all analyses. Taxa displayed on the right: a. *Cyrotyphus vestitus* (Agnesiotidini) (photo credit: Rolf Oberprieler), b. *Pachyura australis* (Pachyurini) (photo credit: Simon Grove), c. *Isacanthodes ganglionicus* (Belini) (photo credit: Rolf Oberprieler), d. *Homalocerus* sp. (Belini) (photo credit: Jeff Gruber), e. *Rhinotia* sp. (Belini) (photo credit: Rolf Oberprieler), f. *Oxycraspedus cribricollis* (Oxycorynini, Oxycraspedina) (photo credit: Adriana Marvaldi), g. *Oxycorynus missionis* (Oxycorynini, Oxycorynina) (photo credit: Adriana Marvaldi), h. *Rhopalotria slossoni* (Oxycorynini, Allocorynina) (photo credit: Shayla Salzman).

The online version of this article includes the following figure supplement(s) for figure 1:

**Figure supplement 1.** Phylogeny estimate of Belidae and nodal test excluding Sanger data.

**Figure supplement 2.** Topology generated by maximum likelihood (ML) analysis of the 46t424g matrix partitioned by gene, with node numbers.

## Results

The 33t423g dataset comprised 97 334 nucleotides, with 37 154 parsimony-informative sites and 31.8% missing data. The 33t424g and 46t424g datasets contained 107 199 nucleotides each, with 45 046 and 45 157 parsimony-informative sites and 33.3% and 51.6% missing data, respectively. These data are summarized in *Supplementary file 1*.

Flanking regions used in the present study are much shorter than those from most ultraconserved elements (9865 bp in this study, averaging 23 bp per locus *versus* 400–1000 bp per locus) and contributed less to missing data (increasing the amount by less than 1.5%). The impact of flanking regions

**Table 1.** Marginal-likelihood estimate (MLE) scores for various BEAST analyses performed for this study, and estimated ages (in Ma) for Belidae crown nodes for each tree prior/clock scheme in BEAST.

Notes: SS, stepping-stone sampling marginal-likelihood estimation; PS, path-sampling marginal-likelihood estimation; median post-burn-in divergence times in millions of years (95% credibility interval).

| Analysis | Tree model | Clock model | MLE SS | MLE PS | Crown Belidae age (Ma) |
|---|---|---|---|---|---|
| A1 | birth-death | 1 ULRC | –142024.6921 | –142025.2950 | 143.8706 [127.5175–159.9967] |
| A2 | Yule | 1 ULRC | –142024.2190 | –142024.9031 | 144.1718 [128.1144–159.9990] |
| A3 | birth-death | 4 ULRC | –135931.3312 | –135932.0701 | 140.7353 [125.4955–157.7755] |
| A4 | Yule | 4 ULRC | –135931.0581 | –135931.8383 | 140.8507 [125.7566–157.9643] |
| **A5** | **birth-death** | **13 ULRC** | **–134225.1645** | **–134225.9867** | **138.4569 [125.5653–154.8667]** |
| A6 | Yule | 13 ULRC | –134226.8380 | –134227.8971 | 138.3267 [125.5619–154.5464] |

on the backbone (tribal-level) relationships has been tested using the reduced datasets (33t423g and 33t424g) with three different partitioning schemes. All six analyses yielded congruent tribal-level relationships and all backbone nodes were robustly supported, except the monophyly of Oxycraspedina + Aglycyderini (*Figure 1—figure supplement 1a*). Analyses of datasets including flanking regions (33t434g) consistently recovered higher statistical support for the sister-group relationship between Oxycraspedina and Aglycyderini, which was further supported by the four-cluster likelihood mapping analyses (*Figure 1—figure supplement 1b, c*). Flanking regions were included in subsequent analyses because this increased the resolution and backbone nodal support values. Backbone topologies generated from the combined dataset (46t424g) were congruent with those from the reduced datasets (*Figure 1*, *Figure 1—figure supplement 1*). The family Belidae, the subfamilies Belinae and Oxycoryninae, the tribes of Belinae (Agnesiotidini, Belini and Pachyurini) and most subtribes of Oxycoryninae (Oxycraspedina, Metrioxenina, Aglycyderina and Oxycorynina) were robustly supported as monophyletic (UFBoot ≥95 AND SH-aLRT ≥80) in the trees resulting from the three partitioning schemes (*Figure 1*). Tribal-level relationships were robustly recovered in Belinae, but generic relationships were only weakly supported even excluding the taxa without anchored-hybrid-enrichment data (33t423g and 33t424g) (*Figure 1—figure supplement 1*). The tribe Oxycorynini in its current concept (*Marvaldi et al., 2006*), was not recovered as a monophylum, Oxycraspedina instead forming the sister-group of Aglycyderini + Metrioxenini, and *Archicorynus* was found to be the sister-group of Allocorynina rather than of all other Oxycoryninae (*Anderson and Marvaldi, 2013*). The phylogenetic positions of Metrioxenini and Archicorynini were unstable across different partitioning schemes (*Figure 1*). This is most likely due to insufficient data, as only Sanger data were generated for representatives of these two tribes. The phylogeny estimate generated with the partition-by-locus scheme was selected as optimal because it closely matches hypotheses based on the analysis of morphological characters (*Marvaldi et al., 2006*). Therefore, the ML tree generated by the 46t424t dataset and partitioned by locus was used for downstream analyses. Details of node supports are summarized in *Figure 1—figure supplement 2* and *Supplementary file 2*.

BEAST analyses with different tree priors/clock schemes yielded similar results, with the crown age of Belidae ranging from 160.0 to 125.6 Ma (*Table 1*). The preferred BEAST analysis, applying a birth-death tree prior with 13 unlinked molecular clocks, was identified using marginal-likelihood estimation (*Table 1*). Divergence-time estimation results revealed an origin of stem Belidae in the Middle Jurassic at 167.8 Ma (95% highest posterior density=185.9–160.0 Ma) and of crown-group Belidae in the early Lower Cretaceous at 138.5 Ma (154.9–125.6 Ma) (*Figure 2*). The highest likelihood among the six biogeographical models tested was DIVALIKE with the unconstrained analysis (M0) (*Supplementary file 3*). Belidae were reconstructed as having a Gondwanan origin, with the two extant subfamilies Belinae and Oxycoryninae originating in the Australian and Neotropical regions, respectively (*Figure 2*).

The most likely ancestral host plants of Belidae were reconstructed as Pinopsida (0.53) or Angiospermae (0.43) (*Figure 1—figure supplement 2*, *Supplementary file 2*). In Belinae, Pinopsida were

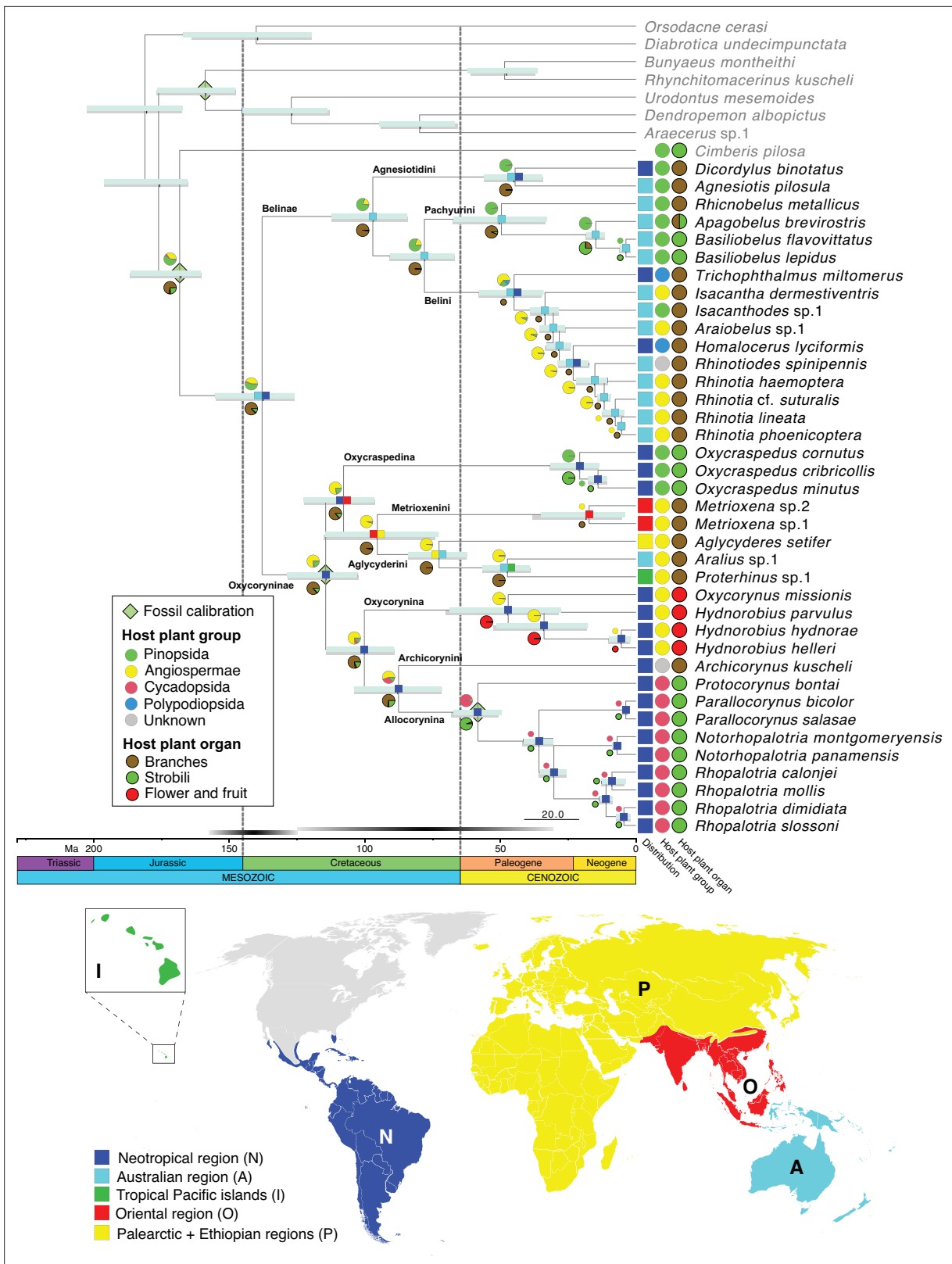

**Figure 2.** Belid timetree. Chronogram was generated using BEAST and a birth-death tree prior with 13 unlinked molecular clocks; pale green diamonds on nodes indicate calibration points. Scale is in Ma. Horizontal bars depict the 95% HPD density of each estimate, cyan bars generated by the birth-death tree prior and gray bars generated by the Yule tree prior. Ancestral area estimation under the likelihood-based implementation of dispersal vicariance analysis and unconstrained dispersal rates (DIVALIKE, M0) performed with BioGeoBEARS. The distribution of each species is mapped to the

*Figure 2 continued*

right of the chronogram. A single most probable ancestral area is mapped at each node. Ancestral state reconstruction of belid larval host plant group and host organ usage under the ER model.

the most likely ancestral host plants (0.79), with a shift to Angiospermae occurring in Belini (0.65) and two independent colonizations of Polypodiopsida (if indeed these are larval hosts and the two genera are not as closely related as indicated by morphology; see *Discussion* below). The ancestral host plants of Oxycoryninae were recovered as Angiospermae (0.76) or Pinopsida (0.21). Extant Oxycraspedina are today associated with Pinopsida and Allocorynina with Cycadopsida (*Figure 2*). Based on the ASR analysis, branches of the host plants were used for larval development by ancestral Belidae with the high possibility (0.87), with a single shift to flowers and fruits (of Angiospermae) in Oxycorynina and at least three shifts to strobili, two to strobili of Pinopsida in Pachyurini and Oxycraspedina and one to (pollen) cones of Cycadopsida in Allocorynina (*Figure 2*).

## Discussion

### Phylogenetic relationships

This study provides the first target-enrichment-based phylogeny estimate of the weevil family Belidae. The monophyly of both subfamilies, Belinae and Oxycoryninae, was recovered, and Aglycyderini were found deeply nested in the latter, supporting the now widely accepted two-subfamily system of belid classification (e.g. *Marvaldi, 2005*). In Belinae, all three tribes (Agnesiotidini, Belini, and Pachyurini) were recovered as monophyletic, supporting the classification of *Kuschel and Leschen, 2003*. Generic relationships in Belini were generally poorly resolved, especially when taxa with Sanger data only were included (*Figure 1—figure supplement 1*, *Figure 1—figure supplement 1*). The placements of *Trichophthalmus* and *Homalocerus* are likely to be artifacts because they result from sparse and non-overlapping data (four Sanger and 13 anchored-hybrid-enrichment loci, respectively). In contrast, morphological data suggest that these two genera are closely related (*Kuschel, 1959*; *Vanin, 1976*; *Kuschel and Leschen, 2003*) and together constitute the sister group of all Australian Belini (Figure 195 of *Kuschel and Leschen, 2003*). In Oxycoryninae, the tribe Oxycorynini in its current concept was found to be polyphyletic, with Oxycraspedina being more closely related to Aglycyderini + Metri-oxenini (*Figure 1*). A close relationship between Aglycyderini and Metrioxenini was also found in the phylogeny estimate derived from morphological characters (*Marvaldi et al., 2006*). All other tribes and the subtribes were recovered as monophyletic groups, in agreement with the results of phylogenies reconstructed from morphological characters by previous authors (*Marvaldi et al., 2006*; *Anderson and Marvaldi, 2013*). However, the placement of *Archicorynus* as a sister-group of Alloco-rynina (albeit with weak support, *Figure 1*) differs from that indicated by morphological characters, which resolved the genus as the sister-group of all other Oxycoryninae (*Anderson and Marvaldi, 2013*).

### Ancestral host plants and larval feeding habits

The larvae of ancestral Belidae were reconstructed as having developed endophytically in branches (not strobili) of their host plants, which is consistent with some previous hypotheses (*Marvaldi et al., 2002*; *Marvaldi, 2005*). Even though our reconstruction provides support for their ancestral hosts most likely being the Pinopsida, which is in accordance with our clade age estimates, angiosperm hosts cannot be completely ruled out, considering that crown angiosperms are now thought to have originated earlier than Belidae, in the interval between the late Permian and the latest Jurassic (256–149 Ma) (*Barba-Montoya et al., 2018*). However, angiosperm families known to be belid hosts, such as Arecaceae, Hydnoraceae, Balanophoraceae, and Fabaceae, did not diversify until the Cretaceous (*Barba-Montoya et al., 2018*; *Li et al., 2019*). During the early evolutionary history of Belidae, conifer lineages were dominant (*Crisp and Cook, 2011*; *Anderson et al., 2007*; *Wang and Ran, 2014*), consistent with Pinopsida being the most likely ancestral host plants of Belidae.

Based on our phylogeny estimate, the host plant shift to Angiospermae may have occurred during the branching of Belini in the Middle to Late Paleogene. Possible host shifts to ferns in the Paleogene in the Neotropical genera *Trichophthalmus* and *Homalocerus* must be viewed with great reservation, as all host records of these genera are only based on the collection of adults, without any evidence of

adult feeding, oviposition or larval development (**Kuschel, 1959**; **Vanin, 1976**). As mentioned above, *Homalocerus* and *Trichophthalmus* might form the earliest-branching lineage of Belini. Therefore, the more plausible scenario for host plant shifts is a single shift to ferns from conifers during the Early to Middle Paleogene, perhaps tracking the radiation of ferns in the late Cretaceous (**Schneider et al., 2004**; **Schuettpelz and Pryer, 2009**).

In Oxycoryninae, the South American genus *Oxycraspedus* (the only extant genus of Oxycraspedina) retains the reconstructed ancestral host association with Pinopsida. Its current hosts, Araucariaceae, were widely distributed in Mesozoic times, and when the oxycorynines diverged by the Lower Cretaceous according to our dated phylogeny estimate, *Araucaria* still constituted an abundant and widely distributed host for beetles (**Sequeira et al., 2000**). It is plausible, then, that the ancestors of the remaining oxycorynine tribes/subtribes were originally associated with these conifers before their decline by the Eocene (**Kershaw and Wagstaff, 2001**) and that these oxycorynine lineages adapted to this decrease in conifer availability by host-shifting to various distantly related taxa of Angiospermae and Cycadopsida.

As is true for weevils generally, the association of Metrioxenini with palms (Arecaceae) may date to the Upper Cretaceous, given our dating and the fossil record of the family (**Marvaldi et al., 2002**; **Matsunaga and Smith, 2021**). However, extant Metrioxenina may only be associated with the genus *Arenga* (**Marvaldi et al., 2006**) — though possible associations with other members of the tribe Caryoteae and the subfamily Coryphoideae have not been explored. The Aglycyderini evidently shifted onto angiosperms as well, but their pattern of host associations is complex and indeterminate as the larvae of most species develop in dead bark and twig tissues of a variety of plants. The larvae of Oxycorynina develop in the flowers and fruits of achlorophyllous root-parasitic plants belonging to two distantly related angiosperm families (Balanophoraceae and Hydnoraceae) (**Ferrer et al., 2011**). The sequence of their host shifts cannot be reconstructed unequivocally; the possibilities include a parallel colonization of these taxa or a shift to Balanophoraceae first and then to Hydnoraceae or vice versa. In any case, our results suggest that these host shifts occurred in the Paleogene at *c.* 47.7 Ma (68.5–27.5 Ma), consistent with the estimated origins of Balanophoraceae and Hydnoraceae (crown age of Hydnoraceae at 54.7 Ma (75–37 Ma) **Naumann et al., 2013**).

All extant species of Allocorynina are known to develop in pollen cones of cycads of the genera *Dioon* and *Zamia* (**Marvaldi et al., 2006**; **O'Brien and Tang, 2015**). Our analysis recovered an initial shift to *Dioon* (hosts of *Parallocorynus* and *Protocorynus*) followed by a subsequent one from *Dioon* to *Zamia* (hosts of *Notorhopalotria* and *Rhopalotria*) (**Figure 2**). The shift to *Dioon* is indicated to have occurred between the late Cretaceous and the early Paleogene, at 58.4 Ma (67.1–49.9 Ma), which is older than the crown age of *Dioon* (24.6–7.5 Ma) but younger than the stem age (207.9–107.0 Ma) (**Condamine et al., 2015**). Australian cycad weevils have also been found to have colonized cycads before their main radiation (**Hsiao et al., 2023**). The cycad tree of life is known for long branches subtending generic-level radiations (**Nagalingum et al., 2011**; **Salas-Leiva et al., 2013**; **Condamine et al., 2015**; **Liu et al., 2022**). The switch from *Dioon* to *Zamia* is estimated to have occurred in the middle Neogene, at 11.0 Ma (13.5–8.6 Ma), which is in line with the crown group age of *Zamia* (9.5 Ma, 22.1–9.0 Ma) (**Calonje et al., 2019**).

There appears to be a pronounced conservatism in the type of tissue consumed by belid larvae, always involving parenchymatous parts of branches or reproductive structures (not pollen or seeds). The type of plant organ used for larval development shows few shifts from the ancestral state (branches): in Belinae to ovulate and/or pollen cones (or 'female and/or male strobili') of Pinopsida (arguably a shift; possibly just opportunistic development alongside that in twigs and branches) in some Agnesiotidini and Pachyurini, in Oxycoryninae to ovulate cones of Pinopsida in *Oxycraspedus*, fleshy flowers/fruits of parasitic angiosperms in Oxycorynina and pollen cones of cycads in Allocorynina (there does not appear to be any feeding on ovulate cones by Allocorynina in nature although they clearly visit them and are presumably capable of feeding on them at least in the short term; **Simon et al., 2023**). As is generally true for plant-feeding beetles (**Farrell and Sequeira, 2004**), host taxon associations are evidently more labile than the use of tissue/organ for larval development. This is particularly noticeable even within some belid genera, whose larvae have been found developing in similar tissues (under bark and in branches) of different plant families, both between congeneric species (e.g. *Sphinctobelus*) and in single species (e.g. *Isacantha*, *Rhicnobelus*); however, there are some cases known in Belidae where the use of plant organs is apparently plastic and opportunistic

(e.g. *Apagobelus brevirostris* (Lea) reared from both stems and cones of *Araucaria*; *Zimmerman, 1994*).

## Palaeogeographical events and host plant shifts

A Gondwanan origin was recovered for Belidae at *c.* 138 Ma, when South America and Australia were connected via Antarctica. The two subfamilies, Belinae and Oxycoryninae, originated and diverged during the separation of East and West Gondwana (*McIntyre et al., 2017*) and both have lineages that still develop on their ancestral host plant. In the East Gondwana clade, Belinae, the divergence time between *Agnesiotis* and *Dicordylus* was estimated at 44.7 Ma (55.9–34.5 Ma), during the Paleocene–Eocene separation of South America and Australia (*Lawver et al., 1992*; *Briggs, 1995*; *McIntyre et al., 2017*). Thus, the evolutionary history of *Dicordylus* might best be explained by persisting in association with the ancestral host plant despite vicariance. The history of the other two South American genera, *Homalocerus* and *Trichophthalmus*, could potentially also be elucidated through the vicariance process (though we cannot rule out the possibility of dispersal to South America). This process may have offered an opportunity for their common ancestor to encounter the new niches of South American ferns, thereby facilitating their shift to new host plants. Such palaeogeographical events shaping host plant usage have also been documented in other phytophagous arthropods (e.g. *Calatayud et al., 2016*).

Due to the lack of Afrocorynina in our sample and the weakly supported placements of Archicorynini and Metrioxenina, the evolutionary history of Oxycoryninae could not be reconstructed with high confidence. Nonetheless, a plausible scenario is that the common ancestor of Aglycyderini +Metrioxenini shifted to angiosperms in the Lower to Middle Cretaceous at 99.9 Ma (113.9–88.9 Ma) (*McIntyre et al., 2017*), diverged in the Ethiopian region by the Middle to Upper Cretaceous and subsequently dispersed to the Palearctic and Oriental regions, the tropical Pacific islands and as far south as New Zealand (Aglycyderina). Additional taxon sampling is needed to illuminate whether this host shift promoted the dispersal of Aglycyderini +Metrioxenini.

## Adaptive radiations in *Rhinotia* and *Proterhinus*

In Belidae, notable taxonomic diversity is observed in the genera *Rhinotia* and *Proterhinus*, with 87 and 168 described species, respectively (*Marvaldi and Ferrer, 2014*; *Brown, 2019*). In contrast, other genera of Belidae comprise fewer than 15 known species (*Marvaldi and Ferrer, 2014*). Host plants of *Rhinotia* are mostly *Acacia* (Fabaceae), the most species-rich plant genus in Australia, with more than 1000 endemic species (*Maslin, 2004*; *Lewis, 2005*). *Acacia* is particularly dominant in arid and semi-arid areas of Australia (*Byrne et al., 2008*). The crown age of *Rhinotia* is here estimated at 11.7 Ma (14.1–9.2 Ma), which postdates the origin of crown *Acacia* (23.9–21 Ma) (*Miller et al., 2013*). The common ancestor of *Rhinotia* might have colonized *Acacia* in the early to middle Miocene when Australia was warm and wet (*Martin, 2006*), and co-diversified during the aridification of Australia from 10 to 6 Ma (*Byrne et al., 2008*). The origin might also have been 20 Ma earlier and involved other now-extinct taxa, with *Rhinotia* being a surviving lineage, or it might have colonized and radiated during the aridification. A similar situation has been postulated for the thrips subfamily Phlaeothripinae, which colonized *Acacia* and diversified into more than 250 species in 35 genera (*McLeish et al., 2007*; *McLeish et al., 2013*).

With 159 species described from the Hawaiian Islands, *Proterhinus* is another example of the spectacular radiation of Hawaiian insects, along with, e.g., the nearly 1000 species of Drosophilidae (Diptera) (*O'Grady et al., 2011*), more than 400 species of *Hyposmocoma* (Lepidoptera: Cosmopterigidae) (*Haines et al., 2014*), over 110 species of *Plagithmysus* (Coleoptera: Cerambycidae) (*Gressitt, 1975*) and more than 190 species of *Nesophrosyne* (Hemiptera: Cicadellidae) (*Bennett and O'Grady, 2013*). Little is known about the host plants of *Proterhinus*, but what is known suggests that host plant ranges (e.g. in *P. deceptor* Perkins, *P. obscurus* Sharp and *P. vestitus* Sharp) are remarkably broader than in other species of Belidae (*Legalov, 2009*). Such host 'jumps' are typical for endemic Hawaiian phytophagous insects, e.g., *Carposina* (Lepidoptera: Carposinidae) (*Medeiros et al., 2016*) and *Nesosydne* (Hemiptera: Delphacidae) (*Roesch Goodman et al., 2012*), but some lineages have high host plant specificity e.g., *Philodoria* (Lepidoptera, Gracillariidae) (*Johns et al., 2018*) and *Nesophrosyne* (Hemiptera: Cicadellidae) (*Bennett and O'Grady, 2013*). Due to limited taxon sampling, we could not estimate the crown age of *Proterhinus* nor when and from where it may have arrived in Hawaii.

## Materials and methods

### Taxon sampling

Forty-six taxa were included in the present study, including eight outgroups representing the chrysomeloid families Chrysomelidae and Orsodacnidae and the weevil families Anthribidae, Cimberididae and Nemonychidae. Ingroup taxon sampling spanned 38 species of Belidae in 24 genera, representing all seven tribes from both subfamilies and 60% of extant belid genera. Genomic data from 33 taxa, 23 newly generated for this study, were used in phylogeny reconstruction. Sanger DNA sequence data (CO1, 16 S, 18 S and 28 S) from six outgroup and 34 ingroup species were also used, including newly generated data for 28 species, and 13 species were represented by Sanger data only (*Supplementary file 4*).

### DNA extraction, library preparation and Illumina DNA sequencing

Total genomic DNA was extracted from the legs, thoracic muscle or the whole body, depending on the size of the specimen, using the G-Biosciences OmniPrep kit (G-Biosciences, Catalog #786–136, St. Louis, MO, U.S.A.), following the manufacturer's protocol, except that samples were incubated for 15 hr instead of 15 min. Final DNA extractions were eluted with 60 µL of nuclease-free water and treated with RNaseA. The remaining body parts were preserved in 95% ethanol as vouchers. Genomic DNA QC statistics were generated for each extracted specimen using a Qubit fluorometer, and DNA quality (fragmentation/degradation and contamination with RNA) was further assessed via gel electrophoresis.

The extracted DNA was fragmented by sonication with a Q800R2 Sonicator (Illumina TruSeq), using 50 µL of the DNA extractions in 0.2 mL strip tubes, targeting a modal fragment size of 350 base pairs. Genomic DNA libraries were constructed using the NEBNext Ultra II DNA Library Prep Kit (NEB #E7645L) with NEBNext Multiplex Oligos for Illumina (Dual Index Primers Sets 1 and 2) (NEB #E7600S and #E7780S), with two-sided size selection around a mode of 480 base pairs. Target enrichment through hybridization followed the myBaits Hybridization Capture for Targeted NGS (Version 5), with 65 °C chosen for the hybridization temperature. We used the published Anchored Hybrid Enrichment Coleoptera Probe set (*Haddad et al., 2018*; *Shin et al., 2018*) and targeted 599 nuclear loci.

Enriched libraries were amplified using KAPA HiFi HotStart ReadyMix. PCR cycling consisted of an initial denaturing step at 98 °C for 2 min, followed by eight cycles of denaturing at 98 °C for 20 s, annealing at 60 °C for 30 s, elongation at 72 °C for 45 s and a final elongation step at 72 °C for 5 min. The 192 enriched and multiplexed libraries were sequenced using 150 bp paired-end reads on an Illumina HiSeq Lane at Novogene Corporation Inc (Sacramento, CA, USA). All raw reads were deposited in the Dryad data repository at https://doi.org/10.5061/dryad.hdr7sqvt7.

### DNA isolation, PCR amplification and Sanger DNA sequencing

DNA extraction and PCR amplification of Sanger data were performed at IADIZA-CONICET (Mendoza, Argentina) and Wellesley College (MA, USA). Total genomic DNA was extracted from adult voucher specimens using an adapted 'salting-out' protocol (*Sunnucks and Hales, 1996*) or the DNeasy Blood and Tissue Kit (QIAGEN, MD, USA). Tissue was processed from one to two legs or part of the thorax. The extracted DNA was stored at –20 °C. Four molecular markers (two nuclear and two mitochondrial) were used in this study: 18 S rDNA (entire), 28 S rDNA (regions D2, D3), 16 S rDNA (regions IV, V) and COI ('barcode' or 5' region). The primers used for amplification and sequencing of the four Sanger loci and PCR conditions are as described by *Marvaldi et al., 2018*. The PCR products were purified and bi-directionally sequenced with the Sanger method, using the Sequencing Service of 'Unidad de Genómica de INTA-Castelar' (Buenos Aires, Argentina) and, in Wellesley, using an ABI PRISM 3100 Genetic Analyzer (Applied Biosystems, Foster City, CA, USA). Electropherograms were edited and contig-assembled using ProSeq v.2.91 (*Filatov, 2002*) and sometimes Sequencher v.5 (Gene-Codes Corp.). All new sequences were deposited in GenBank under accession numbers PP832953–PP832961 and PP840348–PP840386 (*Supplementary file 4*).

### Sequence assembly and orthology prediction

The dataset preparation procedure for anchored-hybrid-enrichment-targeted loci used is as outlined by *Breinholt et al., 2018*. A reference set was prepared using genomic coding sequences (CDS) from nine coleopteran genomes: *Anoplophora glabripennis* (Motschulsky) (Cerambycidae,

GCA_000390285.2), *Aethina tumida* Murray (Nitidulidae, GCA_001937115.1), *Callosobruchus maculatus* (Fabricius) (Chrysomelidae, GCA_900659725.1), *Dendroctonus ponderosae* (Hopkins) (Curculionidae, GCA_020466585.1), *Diabrotica virgifera* LeConte (Chrysomelidae, GCA_003013835.2), *Gonioctena quinquepunctata* (Fabricius) (Chrysomelidae, GCA_018342105.1), *Leptinotarsa decemlineata* Say (Chrysomelidae, GCA_000500325.2), *Sitophilus oryzae* (Linnaeus) (Curculionidae, GCA_002938485.2), and *Tribolium castaneum* (Herbst) (Tenebrionidae, GCA_000002335.3). Raw reads were assembled using an iterative baited assembly (IBA) after filtering with Trim Galore! v.0.4.0 (bioinformatics.babraham.ac.uk). Orthology was determined using the *T. castaneum* genome as a reference, and single-hit and genome mapping location criteria were used with NCBI Blastn (*Camacho et al., 2009*). Cross-contamination checks were conducted with USEARCH (*Edgar, 2010*), and sequences with >99% identity across different subfamilies were identified and removed. Cleaned sequences were aligned in MAFFT v.7.245 (*Katoh and Standley, 2013*), and isoform consensuses were generated using FASconCAT-G 1.02 (*Kück and Longo, 2014*).

Following the method outlined by *Teasdale et al., 2016* and *Li et al., 2022a*, we used a blast-based method to extract Sanger genes from genomic sequences. In short, raw sequence data were assembled using SOAPdenovo v.2 (*Li et al., 2015*), the four Sanger genes were identified using an all-by-all tBlastx search, reads were mapped on potential orthologous sequences using BBmap v.35.85 (*Bushnell, 2014*), and a final consensus sequence was generated after variants were called using GATK v.4.1.1.0 (*McKenna et al., 2010*). Extracted sequences were compared with Sanger sequencing results, and sequences with higher quality and longer reads were maintained for subsequent phylogenetic analysis. The sequences of the ribosomal markers (nuclear 18 S and 28 S and mitochondrial 16 S) were aligned using information on the secondary structure of the arthropod rRNA genes to identify homologous positions as well as regions of ambiguous alignment to be excluded from analyses (*Gillespie et al., 2006*; *Marvaldi et al., 2009*; *Marvaldi et al., 2018*).

## Dataset preparation

Anchored-hybrid-enrichment probe sets comprise highly conserved coding probe regions (i.e. exons) and more variable, generally, non-coding flanking regions (e.g. introns or intergenic regions) located on flanks of the probe region (*Lemmon et al., 2012*; *Haddad et al., 2018*; *Shin et al., 2018*). Following the pipeline, we trimmed flanking regions with 1.5 entropy and 50% density cutoffs at each site in the nucleotide sequence alignments (*Breinholt et al., 2018*). AliView v1.18 (*Larsson, 2014*) was used to manually check each nucleotide alignment to separate 'flanks' from 'probe regions' and ensure that the probe region was in the correct open reading frame (ORF). We used a long-branch detection protocol to investigate the possibility of external contamination, paralogous sequences, and significant sequencing/assembly errors (longbranchpruner.pl available on Osiris, http://galaxy-dev.cnsi.ucsb.edu/osiris/). We produced maximum-likelihood (ML) gene trees from nucleotide (NT)-probe-region multiple-sequence alignments (MSAs) in IQ-TREE v.2.0.6 (*Nguyen et al., 2015*), conducting a full model test for each gene. We pruned tip sequences that exceeded eight standard deviations from the mean tip length of the gene tree from NT MSAs. Loci with <40% taxon coverage were excluded.

Five hundred eighty loci were assembled across 35 taxa, and 419 loci were selected for phylogenetic inference. Four Sanger genes were included, and flanking regions were concatenated and treated as a single locus. Cleaned MSAs were concatenated using Phyx v.1.1 (*Brown et al., 2017*) to generate the dataset with 46 taxa and 424 loci for phylogenetic inference (46t424g). The 13 taxa represented only by Sanger data were excluded to evaluate the impact of missing data and flanking regions, generating two datasets, 33t424g with 33 taxa and 424 loci and 33t423g with 33 taxa and 423 loci (flanking regions excluded).

## Phylogenetic analyses and tests of node support

We conducted ML phylogenetic analyses in IQ-TREE v2.1.3 (*Nguyen et al., 2015*). For the 33t423g and 33t424g datasets, three partitioning schemes were used: (1) unpartitioned, (2) partitioned by locus, (3) partitioned by locus and codon position. All schemes were model-tested using ModelFinder (*Kalyaanamoorthy et al., 2017*) as implemented in IQ-TREE. The best partitioning scheme for the 33t423g dataset was found after merging possible partitions (using the '-MFP+MERGE' command) and determining the best scheme under the Bayesian information criterion (BIC). For the 33t424g and 46t424g datasets, the best schemes of the 33t423g dataset were used with the GTR + ASC model

added for the flanking regions. An initial 1000 parsimony trees were generated in IQ-TREE with the command '-ninit 1000', and 100 trees with the fewest steps were used to initialize the candidate set (-ntop 100), considering all possible nearest-neighbor interchanges (-allnni). These 100 trees were maintained in the candidate set during the ML tree search (-nbest 100), and unsuccessful runs were terminated after 1000 iterations (-nstop 1000). Perturbation strength was set to 0.2 (-pers 0.2), as recommended for datasets with many short sequences. We used nearest-neighbor interchange (NNI) branch swapping to improve the tree search and limit overestimating branch supports due to severe model violations ('-bnni' command). Node supports were computed with 1000 UFBoot replicates ('-B' command) (*Minh et al., 2013*; *Hoang et al., 2018*) and SH-aLRT ('-alrt' command) (*Guindon et al., 2010*).

Both concatenation and gene coalescence approaches were used for tree estimation on dataset 46t424g. For concatenated analyses, partitioning schemes and parameters used were the same as those for the 33t424g and 33t423g datasets. Nodes were classified as 'robustly supported' when they were recovered with support values of UFBoot ≥95 AND SH-aLRT ≥80, as 'moderately supported' when UFBoot ≥95 OR SH-aLRT ≥80 and as 'weakly supported' when UFBoot <95 AND SH-aLRT <80 (*Minh et al., 2013*; *Hoang et al., 2018*).

The best evolutionary model for coalescence analyses was found using ModelFinder Plus ('-MFP' command) for each gene, followed by likelihood tree searches using a partitioning scheme generated in IQ-TREE. We applied nearest-neighbor interchange (NNI) branch swapping to improve the tree search and limit overestimating branch supports due to severe model violations ('-bnni' command). Nodal support was computed with 1000 UFBoot replicates ('-B' command) (*Minh et al., 2013*; *Hoang et al., 2018*). Multi-species-coalescent (MSC) analyses based on these single trees were calculated using ASTRAL-III v5.6.2 (*Zhang et al., 2018*). Nodes were classified as 'robustly supported' when recovered with support values ≥0.95.

Four-cluster likelihood mapping (FcLM; *Strimmer and von Haeseler, 1997*) was performed in IQ-TREE using the 33t423g and 33t424g datasets and ModelFinder-determined partitions (partition by locus and codon position) to quantify support for the sister relationship between Allocorynina and Oxycraspedina. The four schemes used to define the four-taxon clusters are Allocorynina, Aglycyderini, Oxycraspedina, and outgroups.

Quartet sampling (*Pease et al., 2018*) provided an alternate examination of branch support. Quartet sampling of internal node scores included a set of three scores: quartet concordance (QC: values near 1 being more concordant and near –1 more discordant), quartet differential (QD: the more equal the frequencies of discordant topologies, the closer to 1; 0 indicating that only one other discordant topology was found) and quartet informativeness (QI: 1 for all replicates informative, 0 for no replicates informative) (*Pease et al., 2018*).

Concordance and disagreement among genes on the selected ML tree generated by the 46t424g dataset were estimated using the gene concordance factor (gCF) implemented in IQ-TREE (*Minh et al., 2020*), as concatenated analyses can return well-supported trees even when the level of gene incongruence is high (e.g. *Jeffroy et al., 2006*; *Kumar et al., 2012*). Nodes were classified as 'robustly supported' when gCF was higher than gene discordance factor gDF1 and gDF2, 'weakly supported' when gCF equaled gDF1 or gDF2 or both and 'not supported' when gCF was lower than gDF1 or gDF2 (more loci supporting an alternative topology).

## Divergence time estimation

Divergence times were estimated in a Bayesian framework using BEAST v1.10.4 (*Suchard et al., 2018*) on the high-performance computing clusters at the University of Memphis and the China Agricultural University. SortaDate (*Smith et al., 2018*) was used to reduce the nucleotide alignment to a computationally tractable matrix (50 loci) using combined results of clock-likeness (get_var_length.py) and least topological conflict with the species tree (get_bp_genetrees.py). Four Sanger genes were included in the BEAST analyses to ensure that all taxa were present in the reduced matrix, regardless of the SortaDate result. Three different initial partitioning strategies were used: (1) unpartitioned; (2) four partitions (three codon positions for protein-coding genes and one partition for non-protein coding genes); (3) 144 partitions by locus and nucleotide codon position. The best-fit model and partitioning schemes were selected by PartitionFinder2 (*Lanfear et al., 2017*) using the greedy algorithm and the BIC scores across all models included in BEAST (models=beast). The reduced concatenated

data matrix was imported into BEAUTi (*Drummond et al., 2012*). Substitution and clock models were unlinked among partitions, and tree models were linked. An uncorrelated relaxed-molecular-clock model (*Drummond et al., 2006*) and a lognormal prior were applied, two tree priors were tested for each partitioning scheme, Yule (pure birth) and birth–death, and a fixed cladogram based on the topology generated by the concatenated ML analysis was used.

Four internal nodes and the root were constrained for calibration based on the fossil record of Belidae. Several supposed belid fossils have been described, but only a few could be confidently placed based on evident synapomorphies. After a careful examination following *Parham et al., 2012*, five fossils were selected for calibration of four internal nodes: (1) *Sinoeuglypheus daohugouensis* Yu, Davis & Shih (*Yu et al., 2019*) (together with other undescribed Daohugou specimens) for the stem of Cimberididae, with a minimum age of 160 Ma based on the age of the boundary between the Oxfordian and Callovian (ICC 2023); (2) *Talbragarus averyi* Oberprieler & Oberprieler (*Oberprieler and Oberprieler, 2012*) for the stem of Rhinorhynchinae, with a minimum age of 147.3 Ma based on analysis of zircons (*Bean, 2006*); (3) *Preclarusbelus vanini* Santos, Mermudes & Fonseca (*Santos et al., 2007*) and *Cratonemonyx martinsnetoi* Legalov (*Gratshev and Legalov, 2014*) for the stem of Oxycoryninae, with a minimum age of 113 Ma based on the Crato Formation as Upper Aptian following *Santos et al., 2011*; (4) *Pleurambus strongylus* Poinar & Legalov for the stem of Allocorynini, with a minimum age of 15 Ma (*Iturralde-Vinent and MacPhee, 1996*; *Iturralde-Vinent and MacPhee, 2019*). Fossil calibrations were introduced as minimum ages of uniform priors, and the lower margin of the estimated timing of origin of Phytophaga (195 Ma) (*McKenna et al., 2019*) was used as a hard maximum age constraint on the calibrated nodes.

Three independent analyses of each clock scheme and tree-prior combination were run to check for convergence. We evaluated the convergence and mixing of MCMC chains in Tracer version 1.6 (*Rambaut et al., 2018*) to ensure that the effective sample sizes (ESS) exceeded 200. The resulting tree files were combined and resampled with a frequency of 100,000 in LogCombiner (BEAST package) and a burn-in of 30%. Subsampled trees were summarized as a maximum-clade-credibility tree using TreeAnnotator (*Rambaut and Drummond, 2015*), with median heights as node heights. Path sampling and stepping-stone sampling (*Xie et al., 2011*; *Baele et al., 2012*; *Baele et al., 2013*) were performed as part of all BEAST analyses to identify the best tree prior and clock scheme combination.

## Ancestral host plant reconstructions

Ancestral reconstruction analyses were performed for two host plant characters on the dated phylogeny: (i) host plant higher taxa (four states): Angiospermae, Cycadopsida, Pinopsida, and Polypodiopsida; (ii) host plant organ (three states): branches, strobili and flower and fruit (*Supplementary file 4*). Stochastic character mapping was conducted with the 'make.simmap' command in the R package phytools (*Revell, 2012*) with 1000 simulations. Character state transitions were assumed to have equal rates ('ER' option).

## Ancestral area estimation

We recognize five bioregions that best account for the distribution of the sampled species in Belidae: Neotropical region (N), Australian region (A), tropical Pacific islands (I), Oriental region (O) and Palearctic + Ethiopian regions (P) (*Supplementary file 4*). We performed an event-based likelihood-ancestral-area estimation using BioGeoBEARS (*Matzke, 2014*). Three models were used: (1) DEC (Dispersal Extinction Cladogenesis; *Ree and Smith, 2008*), (2) DIVALIKE (a likelihood-based implementation of dispersal vicariance analysis, originally parsimony-based; *Ronquist, 1997*), (3) BAYAREALIKE (a likelihood implementation of BayArea, originally Bayesian; *Landis et al., 2013*). All models were also evaluated under a constrained analysis (M1), in which we considered palaeogeographical events that occurred in the past 160 Ma over four-time slices (160–125 Ma, 125–35 Ma, 35–28 Ma, 28 Ma to present); sliced by three events: separation of East and West Gondwana, separation of South America and Australia (*McIntyre et al., 2017*), origin of Hawaii islands (*Price and Clague, 2002*) and geographical distance variation for a total of six scenarios. The maximum allowed ancestral area was restricted to two. Time slices for geographical events used were those of *McIntyre et al., 2017*. The Akaike Information Criterion (AIC *Burnham and Anderson, 1998*) and the corrected Akaike Information Criterion (AICc *Burnham and Anderson, 2002*) were calculated to select the best-fitting model.

## Acknowledgements

This study was funded by grants from the United States National Science Foundation DEB:1355169 and 2110053 to DDM. XL was supported by the 2115 Talent Development Program of China Agricultural University. AEM acknowledges support by CONICET through grant PIP 3108. The authors acknowledge the China Agricultural University and the University of Memphis for providing high-performance computing platforms and support that contributed to the research results reported in this study.

## Additional information

### Funding

| Funder | Grant reference number | Author |
|---|---|---|
| National Science Foundation | DEB:1355169 | Duane D McKenna |
| National Science Foundation | 2110053 | Duane D McKenna |
| China Agricultural University | 2115 Talent Development Program | Xuankun Li |
| Consejo Nacional de Investigaciones Científicas y Técnicas | PIP 3108 | Adriana E Marvaldi |

The funders had no role in study design, data collection and interpretation, or the decision to submit the work for publication.

### Author contributions

Xuankun Li, Data curation, Software, Formal analysis, Investigation, Visualization, Methodology, Writing – original draft, Writing – review and editing; Adriana E Marvaldi, Conceptualization, Resources, Validation, Investigation, Writing – review and editing; Rolf G Oberprieler, Resources, Validation, Investigation, Writing – review and editing; Dave Clarke, Brian D Farrell, Andrea Sequeira, M Silvia Ferrer, Charles O'Brien, Shayla Salzman, William Tang, Validation, Writing – review and editing; Seunggwan Shin, Software, Writing – review and editing; Duane D McKenna, Conceptualization, Resources, Supervision, Funding acquisition, Validation, Investigation, Visualization, Methodology, Writing – original draft, Project administration, Writing – review and editing

### Author ORCIDs

Xuankun Li ⬥ https://orcid.org/0000-0002-0622-2064
Andrea Sequeira ⬥ http://orcid.org/0000-0001-6771-2991
Shayla Salzman ⬥ https://orcid.org/0000-0001-6808-7542
Duane D McKenna ⬥ https://orcid.org/0000-0002-7823-8727

Reviewer #1 (Public review): https://doi.org/10.7554/eLife.97552.3.sa1
Reviewer #2 (Public review): https://doi.org/10.7554/eLife.97552.3.sa2
Author response https://doi.org/10.7554/eLife.97552.3.sa3

## Additional files

### Supplementary files

- Supplementary file 1. Summary of matrices analyzed.
- Supplementary file 2. Summary of node support generated by concatenated ML analysis, coalescence approach, gene concordance factors (gCF), and Quartet Sampling; the posterior possibility of characters generated by ancestral state reconstructions.
- Supplementary file 3. Results of the BioGeoBEARS analyses.

• Supplementary file 4. Taxa sampled in the present study, with the number of loci retained after all cleaning steps, host plant group and organ coding schemes, and distribution coding schemes.

• MDAR checklist

## 1 Data availability

All new Sanger sequences were deposited in GenBank under accession numbers PP832953-PP832961 and PP840348-PP840386.

The following datasets were generated:

| Author(s) | Year | Dataset title | Dataset URL | Database and Identifier |
|---|---|---|---|---|
| Li X, Marvaldi AE, Oberprieler RG, Clarke D, Farrell BD, Sequeira A, Ferrer MS, O'Brien C, Salzman S, Shin S, Tang W, McKenna DD | 2024 | Data from: The evolutionary history of the ancient weevil family Belidae (Coleoptera: Curculionoidea) reveals the marks of Gondwana breakup and major floristic turnovers, including the rise of angiosperms | https://datadryad. org/stash/dataset/ doi:10.5061/dryad. hdr7sqvt7 | Dryad Digital Repository, 10.5061/dryad.hdr7sqvt7 |
| Li X, Marvaldi AE, Oberprieler RG, Clarke D, Farrell BD, Sequeira A, Ferrer MS, OBrien C, Salzman S, Shin S, Tang W, McKenna D | 2024 | Rhicnobelus metallicus cytochrome c oxidase subunit I (COX1) gene, partial cds; mitochondrial | https://www.ncbi. nlm.nih.gov/nuccore/ PP832953 | NCBI GenBank, PP832953 |
| Li X, Marvaldi AE, Oberprieler RG, Clarke D, Farrell BD, Sequeira A, Ferrer MS, OBrien C, Salzman S, Shin S, Tang W, McKenna D | 2024 | Agnesiotis pilosula cytochrome c oxidase subunit I (COX1) gene, partial cds; mitochondrial | https://www.ncbi. nlm.nih.gov/nuccore/ PP832954 | NCBI GenBank, PP832954 |
| Li X, Marvaldi AE, Oberprieler RG, Clarke D, Farrell BD, Sequeira A, Ferrer MS, OBrien C, Salzman S, Shin S, Tang W, McKenna D | 2024 | Metrioxena sp. 2 AEM-2024a cytochrome c oxidase subunit I (COX1) gene, partial cds; mitochondrial | https://www.ncbi. nlm.nih.gov/nuccore/ PP832955 | NCBI GenBank, PP832955 |
| Li X, Marvaldi AE, Oberprieler RG, Clarke D, Farrell BD, Sequeira A, Ferrer MS, OBrien C, Salzman S, Shin S, Tang W, McKenna D | 2024 | Oxycraspedus cornutus cytochrome c oxidase subunit I (COX1) gene, partial cds; mitochondrial | https://www.ncbi. nlm.nih.gov/nuccore/ PP832956 | NCBI GenBank, PP832956 |
| Li X, Marvaldi AE, Oberprieler RG, Clarke D, Farrell BD, Sequeira A, Ferrer MS, OBrien C, Salzman S, Shin S, Tang W, McKenna D | 2024 | Oxycraspedus minutus cytochrome c oxidase subunit I (COX1) gene, partial cds; mitochondrial | https://www.ncbi. nlm.nih.gov/nuccore/ PP832957 | NCBI GenBank, PP832957 |
| Li X, Marvaldi AE, Oberprieler RG, Clarke D, Farrell BD, Sequeira A, Ferrer MS, OBrien C, Salzman S, Shin S, Tang W, McKenna D | 2024 | Hydnorobius hydnorae cytochrome c oxidase subunit I (COX1) gene, partial cds; mitochondrial | https://www.ncbi. nlm.nih.gov/nuccore/ PP832958 | NCBI GenBank, PP832958 |

*Continued on next page*

*Continued*

| Author(s) | Year | Dataset title | Dataset URL | Database and Identifier |
|---|---|---|---|---|
| Li X, Marvaldi AE, Oberprieler RG, Clarke D, Farrell BD, Sequeira A, Ferrer MS, OBrien C, Salzman S, Shin S, Tang W, McKenna D | 2024 | Hydnorobius helleri cytochrome c oxidase subunit I (COX1) gene, partial cds; mitochondrial | https://www.ncbi.nlm.nih.gov/nuccore/PP832959 | NCBI GenBank, PP832959 |
| Li X, Marvaldi AE, Oberprieler RG, Clarke D, Farrell BD, Sequeira A, Ferrer MS, OBrien C, Salzman S, Shin S, Tang W, McKenna D | 2024 | Rhopalotria slossonae cytochrome c oxidase subunit I (COX1) gene, partial cds; mitochondrial | https://www.ncbi.nlm.nih.gov/nuccore/PP832960 | NCBI GenBank, PP832960 |
| Li X, Marvaldi AE, Oberprieler RG, Clarke D, Farrell BD, Sequeira A, Ferrer MS, OBrien C, Salzman S, Shin S, Tang W, McKenna D | 2024 | Archicorynus kuscheli cytochrome c oxidase subunit I (COX1) gene, partial cds; mitochondrial | https://www.ncbi.nlm.nih.gov/nuccore/PP832961 | NCBI GenBank, PP832961 |
| Li X, Marvaldi AE, Oberprieler RG, Clarke D, Farrell BD, Sequeira A, Ferrer MS, OBrien C, Salzman S, Shin S, Tang W, McKenna D | 2024 | Apagobelus brevirostris voucher 0114 small subunit ribosomal RNA gene, partial sequence | https://www.ncbi.nlm.nih.gov/nuccore/PP840348 | NCBI GenBank, PP840348 |
| Li X, Marvaldi AE, Oberprieler RG, Clarke D, Farrell BD, Sequeira A, Ferrer MS, OBrien C, Salzman S, Shin S, Tang W, McKenna D | 2024 | Basiliobelus lepidus voucher 0125 small subunit ribosomal RNA gene, partial sequence | https://www.ncbi.nlm.nih.gov/nuccore/PP840349 | NCBI GenBank, PP840349 |
| Li X, Marvaldi AE, Oberprieler RG, Clarke D, Farrell BD, Sequeira A, Ferrer MS, OBrien C, Salzman S, Shin S, Tang W, McKenna D | 2024 | Rhicnobelus metallicus voucher BEPARh01 small subunit ribosomal RNA gene, partial sequence | https://www.ncbi.nlm.nih.gov/nuccore/PP840350 | NCBI GenBank, PP840350 |
| Li X, Marvaldi AE, Oberprieler RG, Clarke D, Farrell BD, Sequeira A, Ferrer MS, OBrien C, Salzman S, Shin S, Tang W, McKenna D | 2024 | Agnesiotis pilosula voucher BEAGAg01 small subunit ribosomal RNA gene, partial sequence | https://www.ncbi.nlm.nih.gov/nuccore/PP840351 | NCBI GenBank, PP840351 |
| Li X, Marvaldi AE, Oberprieler RG, Clarke D, Farrell BD, Sequeira A, Ferrer MS, OBrien C, Salzman S, Shin S, Tang W, McKenna D | 2024 | Araiobelus filum voucher BEBEAr01 small subunit ribosomal RNA gene, partial sequence | https://www.ncbi.nlm.nih.gov/nuccore/PP840352 | NCBI GenBank, PP840352 |

*Continued on next page*

*Continued*

| Author(s) | Year | Dataset title | Dataset URL | Database and Identifier |
|---|---|---|---|---|
| Li X, Marvaldi AE, Oberprieler RG, Clarke D, Farrell BD, Sequeira A, Ferrer MS, OBrien C, Salzman S, Shin S, Tang W, McKenna D | 2024 | Trichophthalmus miltomerus voucher BETR01 small subunit ribosomal RNA gene, partial sequence | https://www.ncbi.nlm.nih.gov/nuccore/PP840353 | NCBI GenBank, PP840353 |
| Li X, Marvaldi AE, Oberprieler RG, Clarke D, Farrell BD, Sequeira A, Ferrer MS, OBrien C, Salzman S, Shin S, Tang W, McKenna D | 2024 | Oxycraspedus minutus voucher OXMI01 small subunit ribosomal RNA gene, partial sequence | https://www.ncbi.nlm.nih.gov/nuccore/PP840354 | NCBI GenBank, PP840354 |
| Li X, Marvaldi AE, Oberprieler RG, Clarke D, Farrell BD, Sequeira A, Ferrer MS, OBrien C, Salzman S, Shin S, Tang W, McKenna D | 2024 | Oxycorynus missionis voucher OXOX01 small subunit ribosomal RNA gene, partial sequence | https://www.ncbi.nlm.nih.gov/nuccore/PP840355 | NCBI GenBank, PP840355 |
| Li X, Marvaldi AE, Oberprieler RG, Clarke D, Farrell BD, Sequeira A, Ferrer MS, OBrien C, Salzman S, Shin S, Tang W, McKenna D | 2024 | Hydnorobius hydnorae voucher OXHY01 small subunit ribosomal RNA gene, partial sequence | https://www.ncbi.nlm.nih.gov/nuccore/PP840356 | NCBI GenBank, PP840356 |
| Li X, Marvaldi AE, Oberprieler RG, Clarke D, Farrell BD, Sequeira A, Ferrer MS, OBrien C, Salzman S, Shin S, Tang W, McKenna D | 2024 | Hydnorobius helleri voucher ER02 small subunit ribosomal RNA gene, partial sequence | https://www.ncbi.nlm.nih.gov/nuccore/PP840357 | NCBI GenBank, PP840357 |
| Li X, Marvaldi AE, Oberprieler RG, Clarke D, Farrell BD, Sequeira A, Ferrer MS, OBrien C, Salzman S, Shin S, Tang W, McKenna D | 2024 | Hydnorobius parvulus voucher MEZ02 small subunit ribosomal RNA gene, partial sequence | https://www.ncbi.nlm.nih.gov/nuccore/PP840358 | NCBI GenBank, PP840358 |
| Li X, Marvaldi AE, Oberprieler RG, Clarke D, Farrell BD, Sequeira A, Ferrer MS, OBrien C, Salzman S, Shin S, Tang W, McKenna D | 2024 | Rhopalotria slossonae voucher OXRhs01 small subunit ribosomal RNA gene, partial sequence | https://www.ncbi.nlm.nih.gov/nuccore/PP840359 | NCBI GenBank, PP840359 |
| Li X, Marvaldi AE, Oberprieler RG, Clarke D, Farrell BD, Sequeira A, Ferrer MS, OBrien C, Salzman S, Shin S, Tang W, McKenna D | 2024 | Archicorynus kuscheli voucher OXArk01 small subunit ribosomal RNA gene, partial sequence | https://www.ncbi.nlm.nih.gov/nuccore/PP840360 | NCBI GenBank, PP840360 |

*Continued on next page*

*Continued*

| Author(s) | Year | Dataset title | Dataset URL | Database and Identifier |
|---|---|---|---|---|
| Li X, Marvaldi AE, Oberprieler RG, Clarke D, Farrell BD, Sequeira A, Ferrer MS, OBrien C, Salzman S, Shin S, Tang W, McKenna D | 2024 | Apagobelus brevirostris voucher 0114 large subunit ribosomal RNA gene, partial sequence | https://www.ncbi.nlm.nih.gov/nuccore/PP840361 | NCBI GenBank, PP840361 |
| Li X, Marvaldi AE, Oberprieler RG, Clarke D, Farrell BD, Sequeira A, Ferrer MS, OBrien C, Salzman S, Shin S, Tang W, McKenna D | 2024 | Basiliobelus lepidus voucher 0125 large subunit ribosomal RNA gene, partial sequence | https://www.ncbi.nlm.nih.gov/nuccore/PP840362 | NCBI GenBank, PP840362 |
| Li X, Marvaldi AE, Oberprieler RG, Clarke D, Farrell BD, Sequeira A, Ferrer MS, OBrien C, Salzman S, Shin S, Tang W, McKenna D | 2024 | Rhicnobelus metallicus voucher BEPARh01 large subunit ribosomal RNA gene, partial sequence | https://www.ncbi.nlm.nih.gov/nuccore/PP840363 | NCBI GenBank, PP840363 |
| Li X, Marvaldi AE, Oberprieler RG, Clarke D, Farrell BD, Sequeira A, Ferrer MS, OBrien C, Salzman S, Shin S, Tang W, McKenna D | 2024 | Agnesiotis pilosula voucher BEAGAg01 large subunit ribosomal RNA gene, partial sequence | https://www.ncbi.nlm.nih.gov/nuccore/PP840364 | NCBI GenBank, PP840364 |
| Li X, Marvaldi AE, Oberprieler RG, Clarke D, Farrell BD, Sequeira A, Ferrer MS, OBrien C, Salzman S, Shin S, Tang W, McKenna D | 2024 | Araiobelus filum voucher BEBEAr01 large subunit ribosomal RNA gene, partial sequence | https://www.ncbi.nlm.nih.gov/nuccore/PP840365 | NCBI GenBank, PP840365 |
| Li X, Marvaldi AE, Oberprieler RG, Clarke D, Farrell BD, Sequeira A, Ferrer MS, OBrien C, Salzman S, Shin S, Tang W, McKenna D | 2024 | Trichophthalmus miltomerus voucher BETR01 large subunit ribosomal RNA gene, partial sequence | https://www.ncbi.nlm.nih.gov/nuccore/PP840366 | NCBI GenBank, PP840366 |
| Li X, Marvaldi AE, Oberprieler RG, Clarke D, Farrell BD, Sequeira A, Ferrer MS, OBrien C, Salzman S, Shin S, Tang W, McKenna D | 2024 | Oxycraspedus cornutus voucher OXCO01 large subunit ribosomal RNA gene, partial sequence | https://www.ncbi.nlm.nih.gov/nuccore/PP840367 | NCBI GenBank, PP840367 |
| Li X, Marvaldi AE, Oberprieler RG, Clarke D, Farrell BD, Sequeira A, Ferrer MS, OBrien C, Salzman S, Shin S, Tang W, McKenna D | 2024 | Oxycraspedus minutus voucher OXMI01 large subunit ribosomal RNA gene, partial sequence | https://www.ncbi.nlm.nih.gov/nuccore/PP840368 | NCBI GenBank, PP840368 |

*Continued on next page*

*Continued*

| Author(s) | Year | Dataset title | Dataset URL | Database and Identifier |
|---|---|---|---|---|
| Li X, Marvaldi AE, Oberprieler RG, Clarke D, Farrell BD, Sequeira A, Ferrer MS, OBrien C, Salzman S, Shin S, Tang W, McKenna D | 2024 | Hydnorobius hydnorae voucher OXHY01 large subunit ribosomal RNA gene, partial sequence | https://www.ncbi.nlm.nih.gov/nuccore/PP840369 | NCBI GenBank, PP840369 |
| Li X, Marvaldi AE, Oberprieler RG, Clarke D, Farrell BD, Sequeira A, Ferrer MS, OBrien C, Salzman S, Shin S, Tang W, McKenna D | 2024 | Hydnorobius helleri voucher ER02 large subunit ribosomal RNA gene, partial sequence | https://www.ncbi.nlm.nih.gov/nuccore/PP840370 | NCBI GenBank, PP840370 |
| Li X, Marvaldi AE, Oberprieler RG, Clarke D, Farrell BD, Sequeira A, Ferrer MS, OBrien C, Salzman S, Shin S, Tang W, McKenna D | 2024 | Hydnorobius parvulus voucher MEZ02 large subunit ribosomal RNA gene, partial sequence | https://www.ncbi.nlm.nih.gov/nuccore/PP840371 | NCBI GenBank, PP840371 |
| Li X, Marvaldi AE, Oberprieler RG, Clarke D, Farrell BD, Sequeira A, Ferrer MS, OBrien C, Salzman S, Shin S, Tang W, McKenna D | 2024 | Rhopalotria slossonae voucher OXRhs01 large subunit ribosomal RNA gene, partial sequence | https://www.ncbi.nlm.nih.gov/nuccore/PP840372 | NCBI GenBank, PP840372 |
| Li X, Marvaldi AE, Oberprieler RG, Clarke D, Farrell BD, Sequeira A, Ferrer MS, OBrien C, Salzman S, Shin S, Tang W, McKenna D | 2024 | Archicorynus kuscheli voucher OXArk01 large subunit ribosomal RNA gene, partial sequence | https://www.ncbi.nlm.nih.gov/nuccore/PP840373 | NCBI GenBank, PP840373 |
| Li X, Marvaldi AE, Marvaldi AE, Clarke D, Farrell BD, Sequeira A, Ferrer MS, OBrien C, Salzman S, Shin S, Tang W, McKenna D | 2024 | Rhicnobelus metallicus voucher BEPARh01 large subunit ribosomal RNA gene, partial sequence; mitochondrial | https://www.ncbi.nlm.nih.gov/nuccore/PP840374 | NCBI GenBank, PP840374 |
| Li X, Marvaldi AE, Oberprieler RG, Clarke D, Farrell BD, Sequeira A, Ferrer MS, OBrien C, Salzman S, Shin S, Tang W, McKenna D | 2024 | Agnesiotis pilosula voucher BEAGAg01 large subunit ribosomal RNA gene, partial sequence; mitochondrial | https://www.ncbi.nlm.nih.gov/nuccore/PP840375 | NCBI GenBank, PP840375 |
| Li X, Marvaldi AE, Oberprieler RG, Clarke D, Farrell BD, Sequeira A, Ferrer MS, OBrien C, Salzman S, Shin S, Tang W, McKenna D | 2024 | Araiobelus filum voucher BEBEAr01 large subunit ribosomal RNA gene, partial sequence; mitochondrial | https://www.ncbi.nlm.nih.gov/nuccore/PP840376 | NCBI GenBank, PP840376 |

*Continued on next page*

*Continued*

| Author(s) | Year | Dataset title | Dataset URL | Database and Identifier |
|---|---|---|---|---|
| Li X, Marvaldi AE, Oberprieler RG, Clarke D, Farrell BD, Sequeira A, Ferrer MS, OBrien C, Salzman S, Shin S, Tang W, McKenna D | 2024 | Trichophthalmus miltomerus voucher BETR01 large subunit ribosomal RNA gene, partial sequence; mitochondrial | https://www.ncbi.nlm.nih.gov/nuccore/PP840377 | NCBI GenBank, PP840377 |
| Li X, Marvaldi AE, Oberprieler RG, Clarke D, Farrell BD, Sequeira A, Ferrer MS, OBrien C, Salzman S, Shin S, Tang W, McKenna D | 2024 | Oxycraspedus cribricollis voucher OXCR01 large subunit ribosomal RNA gene, partial sequence; mitochondrial | https://www.ncbi.nlm.nih.gov/nuccore/PP840378 | NCBI GenBank, PP840378 |
| Li X, Marvaldi AE, Oberprieler RG, Clarke D, Farrell BD, Sequeira A, Ferrer MS, Ferrer MS, OBrien C, Salzman S, Shin S, Tang W, McKenna D | 2024 | Oxycraspedus cornutus voucher OXCO01 large subunit ribosomal RNA gene, partial sequence; mitochondrial | https://www.ncbi.nlm.nih.gov/nuccore/PP840379 | NCBI GenBank, PP840379 |
| Li X, Marvaldi AE, Oberprieler RG, Clarke D, Farrell BD, Sequeira A, Ferrer MS, OBrien C, Salzman S, Shin S, Tang W, McKenna D | 2024 | Oxycraspedus minutus voucher OXMI01 large subunit ribosomal RNA gene, partial sequence; mitochondrial | https://www.ncbi.nlm.nih.gov/nuccore/PP840380 | NCBI GenBank, PP840380 |
| Li X, Marvaldi AE, Oberprieler RG, Clarke D, Farrell BD, Sequeira A, Ferrer MS, OBrien C, Salzman S, Shin S, Tang W, McKenna D | 2024 | Oxycorynus missionis voucher OXOX01 large subunit ribosomal RNA gene, partial sequence; mitochondrial | https://www.ncbi.nlm.nih.gov/nuccore/PP840381 | NCBI GenBank, PP840381 |
| Li X, Marvaldi AE, Oberprieler RG, Clarke D, Farrell BD, Sequeira A, Ferrer MS, OBrien C, Salzman S, Shin S, Tang W, McKenna D | 2024 | Hydnorobius hydnorae voucher OXHY01 large subunit ribosomal RNA gene, partial sequence; mitochondrial | https://www.ncbi.nlm.nih.gov/nuccore/PP840382 | NCBI GenBank, PP840382 |
| Li X, Marvaldi AE, Oberprieler RG, Clarke D, Farrell BD, Sequeira A, Ferrer MS, OBrien C, Salzman S, Shin S, Tang W, McKenna D | 2024 | Hydnorobius helleri voucher ER02 large subunit ribosomal RNA gene, partial sequence; mitochondrial | https://www.ncbi.nlm.nih.gov/nuccore/PP840383 | NCBI GenBank, PP840383 |
| Li X, Marvaldi AE, Oberprieler RG, Clarke D, Farrell BD, Sequeira A, Ferrer MS, OBrien C, Salzman S, Shin S, Tang W, McKenna D | 2024 | Hydnorobius parvulus voucher MEZ02 large subunit ribosomal RNA gene, partial sequence; mitochondrial | https://www.ncbi.nlm.nih.gov/nuccore/PP840384 | NCBI GenBank, PP840384 |

*Continued on next page*

*Continued*

| Author(s) | Year | Dataset title | Dataset URL | Database and Identifier |
|---|---|---|---|---|
| Li X, Marvaldi AE, Oberprieler RG, Clarke D, Farrell BD, Sequeira A, Ferrer MS, OBrien C, Salzman S, Shin S, Tang W, McKenna D | 2024 | Rhopalotria slossonae voucher OXRhs01 large subunit ribosomal RNA gene, partial sequence; mitochondrial | https://www.ncbi.nlm.nih.gov/nuccore/PP840385 | NCBI GenBank, PP840385 |
| Li X, Marvaldi AE, Oberprieler RG, Clarke D, Farrell BD, Sequeira A, Ferrer MS, OBrien C, Salzman S, Shin S, Tang W, McKenna D | 2024 | Archicorynus kuscheli voucher OXArk01 large subunit ribosomal RNA gene, partial sequence; mitochondrial | https://www.ncbi.nlm.nih.gov/nuccore/PP840386 | NCBI GenBank, PP840386 |

The following previously published dataset was used:

| Author(s) | Year | Dataset title | Dataset URL | Database and Identifier |
|---|---|---|---|---|
| Haddad S, Shin S, Lemmon AR, Lemmon EM, Svacha P, Farrell B, Slipinski A, Windsor D, McKenna DD | 2018 | Data from: Anchored hybrid enrichment provides new insights into the phylogeny and evolution of longhorned beetles (Cerambycidae) | https://doi.org/10.5061/dryad.v0b7v | Dryad Digital Repository, 10.5061/dryad.v0b7v |

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
