## [Editor Report · eLife assessment]

Through anchored phylogenomic analyses, this **important** study offers fresh insights into the evolutionary history of the plant diet and geographic distribution of Belidae weevil beetles. Employing robust methodological approaches, the authors propose that certain belid lineages have maintained a continuous association with Araucaria hosts since the Mesozoic era. Although the biogeographical analysis is somewhat limited by uncertainties in vicariance explanations, this **convincing** study enhances our understanding of Belidae's evolutionary dynamics and provides new perspectives on ancient community ecology.

---

## [Referee Report · Reviewer #1 (Public review)]

This is a very nice study of Belidae weevils using anchored phylogenomics that presents a new backbone for the family and explores, despite a limited taxon sampling, several evolutionary aspects of the group. I find that the methodology is appropriate, and all analytical steps are well presented. The paper is well written and presents interesting aspects of Belidae systematics and evolution. The major weakness of the study being the very limited taxon sampling that has deep implications for the discussion of ancestral estimations.

---

## [Referee Report · Reviewer #2 (Public review)]

Summary:

The authors used a combination of anchored hybrid enrichment and Sanger sequencing to construct a phylogenomic data set for the weevil family Belidae. Using evidence from fossils and previous studies they are able to estimate a phylogenetic tree with a range of dates for each node - a timetree. They use this to reconstruct the history of the belids' geographic distributions and associations with their hostplants. They infer that the belids' association with conifers pre-dates the rise of the angiosperms. They offer an interpretation of belid history in terms of the breakup of Gondwanaland, but acknowledge that they cannot rule out alternative interpretations that invoke dispersal.

Strengths:

The strength of any molecular-phylogenetic study hinges on four things: the extent of the sampling of taxa; the extent of the sampling of loci (DNA sequences) per genome; the quality of the analysis; and - most subjectively - the importance and interest of the evolutionary questions the study allows the authors to address. The first two of these, sampling of taxa and loci, impose a tradeoff: with finite resources, do you add more taxa or more loci? The authors follow a reasonable compromise here, obtaining a solid anchored-enrichment phylogenomic data set (423 genes, >97 kpb) for 33 taxa, but also doing additional analyses that included 13 additional taxa from which only Sanger sequencing data from 4 genes was available. The taxon sampling was pretty solid, including all 7 tribes and a majority of genera in the group. The analyses also seemed to be solid - exemplary, even, given the data available.

This leaves the subjective question of how interesting the results are. The very scale of the task that faces systematists in general, and beetle systematists in particular, presents a daunting challenge to the reader's attention: there are so many taxa, and even a sophisticated reader may never have heard of any of them. Thus it's often the case that such studies are ignored by virtually everyone outside a tiny cadre of fellow specialists. The authors of the present study make an unusually strong case for the broader interest and importance of their investigation and of its focal taxon, the belid weevils.

The belids are of special interest because - in a world churning with change and upheaval, geologically and evolutionarily - relatively little seems to have been going on with them, at least with some of them, for the last hundred million years or so. The authors make a good case that the Araucaria-feeding belid lineages found in present-day Australasia and South America have been feeding on Araucaria continuously since the days when it was a dominant tree taxon nearly worldwide, before it was largely replaced by angiosperms. Thus these lineages plausibly offer a modern glimpse of an ancient ecological community.

Comments on current version:

The MS was already in pretty good shape last time around, and the authors have made most of the minor revisions and copy-edits suggested by the reviewers. There may be a few remaining points of disagreement with the reviewers, but these seem to be minor matters of opinion and nothing that ought to delay publication.

---

## [Author Response]

The following is the authors’ response to the original reviews.

**Public Reviews:**

**Reviewer #1 (Public Review):**
This is a very nice study of Belidae weevils using anchored phylogenomics that presents a new backbone for the family and explores, despite a limited taxon sampling, several evolutionary aspects of the group. The phylogeny is useful to understand the relationships between major lineages in this group and preliminary estimation of ancestral traits reveals interesting patterns linked to host-plant diet and geographic range evolution. I find that the methodology is appropriate, and all analytical steps are well presented. The paper is well-written and presents interesting aspects of Belidae systematics and evolution. The major weakness of the study is the very limited taxon sampling which has deep implications for the discussion of ancestral estimations.

Thank you for these comments.

The taxon sampling only appears limited if counting the number of species. However, 70 % of belid species diversity belongs to just two genera. Moreover, patterns of host plant and host organ usage and distribution are highly conserved within genera and even tribes. Therefore, generic-level sampling is a reasonable measure of completeness. Although 60 % of the generic diversity was sampled in our study, we acknowledge that our discussion of ancestral estimations would be stronger if at least one genus of

Afrocorynina and the South American genus of Pachyurini could be included.

**Reviewer #2 (Public Review):**
Summary:The authors used a combination of anchored hybrid enrichment and Sanger sequencing to construct a phylogenomic data set for the weevil family Belidae. Using evidence from fossils and previous studies they can estimate a phylogenetic tree with a range of dates for each node - a time tree. They use this to reconstruct the history of the belids' geographic distributions and associations with their host plants. They infer that the belids' association with conifers pre-dates the rise of the angiosperms. They offer an interpretation of belid history in terms of the breakup of Gondwanaland but acknowledge that they cannot rule out alternative interpretations that invoke dispersal.Strengths:The strength of any molecular-phylogenetic study hinges on four things: the extent of the sampling of taxa; the extent of the sampling of loci (DNA sequences) per genome; the quality of the analysis; and - most subjectively - the importance and interest of the evolutionary questions the study allows the authors to address. The first two of these, sampling of taxa and loci, impose a tradeoff: with finite resources, do you add more taxa or more loci? The authors follow a reasonable compromise here, obtaining a solid anchored-enrichment phylogenomic data set (423 genes, >97 kpb) for 33 taxa, but also doing additional analyses that included 13 additional taxa from which only Sanger sequencing data from 4 genes was available. The taxon sampling was pretty solid, including all 7 tribes and a majority of genera in the group. The analyses also seemed to be solid - exemplary, even, given the data available.This leaves the subjective question of how interesting the results are. The very scale of the task that faces systematists in general, and beetle systematists in particular, presents a daunting challenge to the reader's attention: there are so many taxa, and even a sophisticated reader may never have heard of any of them. Thus it's often the case that such studies are ignored by virtually everyone outside a tiny cadre of fellow specialists. The authors of the present study make an unusually strong case for the broader interest and importance of their investigation and its focal taxon, the belid weevils.The belids are of special interest because - in a world churning with change and upheaval, geologically and evolutionarily - relatively little seems to have been going on with them, at least with some of them, for the last hundred million years or so. The authors make a good case that the Araucaria-feeding belid lineages found in present-day Australasia and South America have been feeding on Araucaria continuously since the days when it was a dominant tree taxon nearly worldwide before it was largely replaced by angiosperms. Thus these lineages plausibly offer a modern glimpse of an ancient ecological community.Weaknesses:I didn't find the biogeographical analysis particularly compelling. The promise of vicariance biogeography for understanding Gondwanan taxa seems to have peaked about 3 or 4 decades ago, and since then almost every classic case has been falsified by improved phylogenetic and fossil evidence. I was hopeful, early in my reading of this article, that it would be a counterexample, showing that yes, vicariance really does explain the history of *something*. But the authors don't make a particularly strong claim for their preferred minimum-dispersal scenario; also they don't deal with the fact that the range of Araucaria was vastly greater in the past and included places like North America. Were there belids in what is now Arizona's petrified forest? It seems likely. Ignoring all of that is methodologically reasonable but doesn't yield anything particularly persuasive.

Thank you for these comments.

The criticism that the biogeographical analysis is “not very compelling” is true to a degree, but it is only a small part of the discussion and, as stated by the reviewer, cannot be made more “persuasive”, in part because of limitations in taxon sampling but also because of uncertainties of host associations (e.g. with ferns). We tried to draw persuasive conclusions while not being too speculative at the same time. Elaborating on our short section here would only make it much more speculative — and dispersal scenarios more so than vicariance ones (at least in Belinae).

**Recommendations for the authors:**

**Reviewer #1 (Recommendations For The Authors):**
I have a few comments relative to this last point of a more general nature:- I think it would be informative in Figure 1 to present family names for the outgroups.

Family names for outgroups have been added to Figure 1.

- There is a summary of matrix composition in the results but I think a table would be better listing all necessary information for each dataset (number of taxa, number of taxa with only Sanger data, parsimony informative sites, GC content, missing data, etc...).

We added Table S4 with detailed information about the matrices.

- Perhaps I missed it, but I didn't find how fossil calibrations were implemented in BEAST (which prior distribution was chosen and with which parameters).

We used uniform priors, this has been added to the Methods section.

- I am worried that the taxon sampling (ca. 10% of the family) is too low to conduct meaningful ancestral estimations, without mentioning the moderately supported relationships among genera and large time credibility intervals. This should be better acknowledged in the paper and perhaps should weigh more into the discussion.

Belidae in general are a rare group of weevils, and it has been a huge effort and a global collaboration to sample all tribes and over 60 % of the generic diversity in the present study. A high degree of conservation of host plant associations, host plant organ usage and distribution are observed within genera and even tribes. Therefore, we feel strongly that the resulting ancestral states are meaningful.

Moreover, 70 % of the belid species diversity belongs to only two genera, *Rhinotia* and *Proterhinus*. Our species sampling is about 36 % if we disregard the 255 species of these two genera.

However, we acknowledge that our results could be improved by sampling more genera of Afrocorynina and Pachyurini. However, these taxa are very hard to collect. We have acknowledged the limitation of our taxon sampling, branching supports and timetree credibility intervals in the discussion to minimize speculative in conclusions.

- It might be nice to have a more detailed discussion of flanking regions. In my experience and from the literature there seems to be increasing concern about the use of these regions in phylogenomic inferences for multiple solid reasons especially the more you go back in time (complex homology assessment, overall gappyness, difficulty to partition the data, etc...)

We tested the impact of flanking regions on the results of our analyses and showed this data did not having a detrimental impact. We added more details about this to the results section of the paper, including information about the cutoffs we used to trim the flanking regions.

**Reviewer #2 (Recommendations For The Authors):**
Line 42, change "recent temporal origins" to "recent origins".

Modified in the text.

Line 97-98, "phylogenetic hypotheses have been proposed for all genera" This is ambiguous. The syntax makes it sound like these were separate hypotheses for each genus - the relationships of the species within them, maybe. However, the context implies that the hypotheses relate to the relationships between the genera. Clarify. "A phylogenetic hypothesis is available for generic relationships in each subfamily. . . " or something.

Modified in the text.

Line 162, ". . . all three subtribes (Agnesiotinidi, Belini. . . " Something's wrong here). Change "subtribes" to "tribes"?

Modified in the text.

Line 219, the comma after "unequivocally" needs to be a semicolon.

Modified in the text.

Line 327 and elsewhere, the abbreviation "AHE" is used but never spelled out; spell out what it stands for at first use. Or why not spell it out every single time? You hardly ever use it and scientists' habit of using lots of obscure abbreviations is a bad one that's worth resisting, especially now that it no longer requires extra ink and paper to spell things out.

Modified in the text.